# Genome-Wide Dissection of Novel QTLs and Genes Associated with Weed Competitiveness in Early-Backcross Selective Introgression-Breeding Populations of Rice (*Oryza sativa* L.)

**DOI:** 10.3390/biology14040413

**Published:** 2025-04-13

**Authors:** Kim Diane Nocito, Varunseelan Murugaiyan, Jauhar Ali, Ambika Pandey, Carlos Casal, Erik Jon De Asis, Niña Gracel Dimaano

**Affiliations:** 1Rice Breeding Platform, International Rice Research Institute, Los Baños 4031, Laguna, Philippines; nocito.kimdiane@gmail.com (K.D.N.); v.murugaiyan@irri.org (V.M.); ambika.pandey46@gmail.com (A.P.); caloy_casal@yahoo.com (C.C.J.); e.deasis@irri.org (E.J.D.A.); 2Institute of Weed Science, Entomology and Plant Pathology, College of Agriculture and Food Science, University of the Philippines Los Baños, Los Baños 4031, Laguna, Philippines; 3Department of Agronomy and Crop Physiology, Institute for Agronomy and Plant Breeding I, Justus Liebig University Giessen, 35390 Giessen, Germany

**Keywords:** direct-seeded rice, weed competitiveness, jungle rice, seedling vigor, single nucleotide polymorphisms, quantitative trait loci, Green Super Rice, candidate genes

## Abstract

The direct-seeded rice (DSR) system, known for its advantages like water conservation and climate adaptability, faces significant yield losses due to weeds, especially jungle rice (*Echinochloa colona*). In this study, we identify genetic traits that improve early seed germination (ESG) and seedling vigor (ESV) in rice, key factors for weed competitiveness. Using a mapping population of 181 breeding lines derived from the Green Super Rice breeding program revealed 19 quantitative trait loci (QTLs), 17 of which are novel, linked to weed competitiveness. These QTLs, located on eight rice chromosomes, highlight candidate genes involved in stress responses. The findings support marker-assisted and genomic selection for breeding weed-competitive rice varieties. This work also emphasizes the potential of gene expression studies, cloning, and CRISPR editing to enhance weed management. By improving rice’s natural weed competitiveness, this research contributes to sustainable DSR farming with reduced reliance on herbicides, increased yield stability, and better resource efficiency.

## 1. Background

Rice is the primary food supply and a significant economic driver throughout most of Asia, providing essential employment and financial stability for rural communities [1,2]. Rice consumption is projected to rise by 30% by 2050, driven by growing populations, urbanization, and consumer preferences, particularly in sub-Saharan Africa, where rice consumption is increasing by over 6% annually [3,4]. The availability of resources necessary for stable rice production, such as land, water, fertilizers, and labor, is diminishing. Additionally, the impact of climate change, marked by higher temperatures and frequent floods and droughts, has led to a significant loss to rice production globally [5]. Recent advancement in technologies like the direct-seeded rice (DSR) farming system is anticipated to emerge as the dominant rice cultivation method, boasting a 15.3% reduction in water usage compared to traditional transplanted-flooded rice fields [6,7]. Furthermore, when combined with short-duration high-yielding inbred and hybrid rice varieties, DSR offers additional benefits such as reduced labor needs, higher yields per unit area, less greenhouse gas emissions, fewer energy inputs, and increased resilience to climate change risks [8]. Despite being viewed as a potential alternative to unsustainable water-intensive transplanted rice systems, the heavy infestation of various weed species limits the large-scale adoption of DSR technology [9].

Weed infestation is one of the most significant constraints in DSR that can lead to severe yield losses. Weed diversity in DSR exhibits a greater diversity of plant species and is more abundant than transplanted rice. As a result, the DSR farming system experiences much higher levels of weed pressure [10]. A global assessment conducted in many rice-producing nations revealed that weed infestation posed a significant biological limitation, impeding rice production [11]. The extent of yield loss in DSR due to weed infestation can vary widely based on several factors, including the weed species present, weed density, timing of weed competition, crop management practices, and environmental conditions. The percentage of yield loss can range from 20% to 90%, depending on the severity of weed infestation and how well weed management practices are implemented [12]. If weeds are not effectively controlled, they can out-compete rice plants for resources such as sunlight, water, and nutrients, leading to stunted growth, yield losses, and poor grain quality [13]. Farmers and agricultural extension services often need to assess local conditions and adopt a combination of cultural, chemical, and mechanical weed control methods to optimize yield and ensure the success of DSR cultivation. Effective weed control in DSR farming systems requires regular monitoring, timely interventions, and the breeding and adoption of weed-competitive rice varieties [14,15].

In DSR farming systems, weed diversity can encompass various species such as barnyardgrass (*Echinochloa crus-galli*), jungle rice (*Echinochloa colona*), broadleaf signalgrass (*Brachiaria platyphylla*), late watergrass (*Echinochloa phyllopogon*), ammannia (*Ammannia* spp.), eclipta (*Eclipta prostrata*), fringerush (*Fimbristylis miliacea*), and other sedges (*Cyperus* spp.). Managing this diversity is crucial for successful DSR establishment, particularly during the rice plant’s most vulnerable early growth stages [10]. Jungle rice is highly competitive and efficient within rice farming systems, particularly in warm and wet environments where C4 plants thrive [16]. Jungle rice’s C4 photosynthetic pathway allows it to efficiently capture and use CO_2_, even under high temperatures and intense sunlight, conditions common in rice paddies [15]. This efficiency enables jungle rice to grow rapidly and outcompete rice, a C3 plant, especially in waterlogged or poorly drained soils typical of rice paddies [17]. Its invasive nature is attributed to its vigorous growth and high seed production, with each plant capable of producing up to 42,000 viable seeds [18]. Its pervasive presence has led to substantial grain yield losses, ranging from 27% to 62%, emphasizing its profound impact on rice production [15]. By elucidating the genetic mechanisms underlying weed competitiveness, our study provides valuable insights for breeding programs to develop rice varieties with improved weed competitiveness.

Weed-competitive rice varieties are distinguished by a suite of traits that collectively confer a competitive advantage against weed pressure [19,20]. These traits encompass early vigor, enabling rapid germination and robust seedling growth, and prolific tillering, resulting in a dense canopy that shades out competing weeds. Tall and erect canopy architecture and broad leaves enhance the shading effect, limiting weed access to sunlight [21]. Favorable root systems, characterized by depth and spread, enable efficient nutrient and water uptake, contributing to the overall vigor of the rice plants [22]. Some varieties also exhibit allelopathic compounds, suppressing weed germination and growth [23]. Genetic factors, including specific genes and quantitative trait loci (QTLs), play a crucial role in defining these traits, facilitating the development of rice varieties with enhanced weed competitiveness [24]. Additionally, traits such as tolerance to weed competition, early maturity, and resistance to diseases further fortify the resilience of these varieties. However, a limited number of QTL studies focused on weed-competitive rice varieties can be attributed to the complexity of weed interactions, which vary significantly across different environments and weed species [25]. Traditionally, research has prioritized yield and resistance to biotic and abiotic stresses over traits specifically enhancing competitiveness against weeds, leading to fewer targeted studies [26]. Additionally, resource limitations and the need for multi-disciplinary approaches in understanding weed competition further hinder extensive QTL mapping efforts in this area [27].

Genes associated with weed-competitive traits in rice play a pivotal role in enhancing the crop’s ability to suppress weed growth. Key genes include *OsPAL4* (Phenylalanine Ammonia Lyase 4), which is crucial for allelopathy by synthesizing natural herbicides that create an unfavorable environment for weed growth [28]. Additionally, *OsERF3* (Ethylene Response Factor 3) modulates ethylene responses, influencing root architecture and elongation, thereby enhancing rice’s competitive ability for below-ground resources [29]. ATP-binding cassette transporters, such as OsABC1-13, contribute to herbicide resistance, providing weed-competitive rice varieties with resilience against standard weed control measures [30]. The *DEP1* (Dense and Erect Panicle 1) gene is associated with increased grain yield and altered plant architecture, which enhances weed competitiveness by promoting a compact growth habit that limits space for weed establishment [31]. While genes such as *OsPAL4*, *OsERF3*, and *DEP1* are believed to be correlated with weed competitiveness in rice due to their roles in allelopathy, root architecture, and plant structure, they are not directly proven to enhance the crop’s ability to suppress weeds. The functional associations of these genes with traits that may influence competitiveness remain largely hypothetical and require further empirical investigation to establish direct links [28,29,31]. Our current study aims to identify key candidate QTLs and genes that will provide valuable insights for targeted breeding programs to create tailored weed-competitive rice varieties suitable for direct-seeded rice (DSR) farming systems.

The Green Super Rice (GSR) project, led by the International Rice Research Institute (IRRI) and the Chinese Academy of Agricultural Sciences (CAASs), successfully developed and released high-yielding and multiple stress-tolerant rice varieties to overcome climate change, resource constraints, and disease pressures [2,32]. One of the GSR project’s primary focuses was enhancing yield and resilience, including the weed competitiveness, which is crucial to sustainable rice cultivation. GSR varieties are tailored to exhibit multiple stress tolerance and competitiveness against weeds through traits like early vigor, prolific tillering, efficient canopy development, and high yielding. Integrating weed-resistant characteristics into elite GSR breeding materials and using DSR technology will ensure rice production’s long-term sustainability and productivity [1,2]. Our current study investigated the genetic and molecular mechanisms underpinning weed competitiveness in rice to expedite the development of weed-competitive varieties for sustainable DSR farming systems. The study’s objectives were to screen for weed-competitiveness in early GSR backcross selective introgression populations of rice and to identify QTLs and candidate genes associated with weed-competitiveness using high-density genome-wide SNP markers.

## 2. Methods

### 2.1. Plant Materials

The elite GSR rice breeding population consists of 181 BC_1_F_6_ early-backcross selective introgression-breeding lines (EB-SILs), which were developed through a cross between the indica rice variety Weed Tolerant Rice1 (WTR1) as the recipient parent, and three donor parents, Haoannong (*japonica*), ChengHui448 (*indica*), and Y134 (*indica*). These breeding populations were advanced using the single seed descent method at the International Rice Research Institute (IRRI) in Los Baños, Philippines. Additional comprehensive information on this breeding process and the development of the population was elucidated in Ali et al. (2018) [33]. Seeds of jungle rice (*Echinochloa colona* L.) were harvested from rice-fallow fields at IRRI during the dry season of 2023. The seeds were then kept in a refrigerator at 4 °C before use in the weed competitiveness screening experiment.

### 2.2. Phenotypic Screening for Weed Competitiveness

Rice seeds were dried in an oven at 60 °C for seven days to disrupt any residual seed dormancy. The dormancy of jungle rice seeds was removed by subjecting the seeds to a 24 h immersion in sterilized water, followed by an additional 24 h incubation period at 40 °C in an oven [28]. The phenotyping of early seed germination (ESG) traits involved the placement of 25 seeds per introgression breeding line along with their parents in a 9 cm diameter Petri dish with wet filter paper, maintaining optimal moisture conditions through regular watering in a germination chamber set at 30 °C with a 12 h photoperiod for 14 days. The set-up was arranged in a complete randomized design (CRD) with 2 replications, leading to a total of 370 Petri dishes, each accommodating up to 25 seeds. Germination parameters like the 2nd and 7th day germination counts, germination rate, coleoptile length, radicle length, and total dry weight, were recorded in the experiment. The Seedling Vigor Index is calculated using the following formula.SVI = Germination Rate (%) × Total Dry Weight (g)

Phenotyping for early seedling vigor (ESV) traits was conducted in a greenhouse with average night and day temperatures of 24 °C and 32 °C, respectively. Rice lines were tested under two conditions: weedy (treatment) and non-weedy (control). The experiment was set up in metal trays (150 cm × 90 cm × 10 cm) filled with sterilized Maahas clay loam soil, using a randomized complete block design (RCBD) with two replications in each condition (weedy and non-weedy), leading to a total of 56 trays, with each tray accommodating up to 13 lines. In the control condition, rice was direct-seeded at a density of 40 seeds per 0.1 m^2^, and non-segregating parent lines (WTR1, Haoannong, ChengHui448, and Y134) were grown as checks in each block. Weed control was ensured through soil sterilization and manual weeding. In the weedy condition, rice was also direct-seeded at the same density (40 seeds per 0.1 m^2^), but with the addition of jungle rice seeds. Both rice and weed seeds were sown together at a depth of 2 cm in a 1:1 ratio to create high weed pressure. Basal fertilizer (60-30-30 kg NPK ha^−1^) was applied in both the control and weedy conditions, following local recommendations. Both setups were watered daily, keeping the soil unsaturated to simulate DSR conditions. Data collection was conducted using five randomly tagged plants per line, excluding border plants. Measurements were taken at 14, 21, and 28 days after sowing (DAS) for parameters such as plant height, leaf count, and tiller number. At 28 DAS, additional traits were recorded, including the seedling vigor index, shoot dry weight, root dry weight, total dry weight, and root length (Appendix A). To minimize any positional effects within the germination chamber and greenhouse, the positions of Petri dishes and metal trays were altered every other day.

### 2.3. Statistical Analysis

A one-way analysis of variance (ANOVA) at a significance level of one percent was used to analyze the phenotypic data for ESG and two-way ANOVA was used for ESV to observe the effects of genotypes and the growing conditions. Pearson’s correlation analysis was conducted to correlate the ESG-related traits among each other, as well as to correlate the ESV-related traits. ANOVA and Pearson’s correlation were performed in R version 4.3 studio (http://www.rstudio.com/, accessed on 8 March 2024), and heat maps were generated using the hmsic package. For ESG, we conducted Principal Component Analysis (PCA) using all measured traits, considering the first two principal components for further analysis. For ESV, we performed PCA using relative trait values to account for proportional differences and biplots were created for the first two components using JMP^®^ (https://www.jmp.com/en_ph/home.html, accessed on 8 March 2024).

### 2.4. SNP Extraction and Physical Map Construction

From the previous publication by Ali et al. (2018) [33], all 181 lines and their source tunable genotyping-by-sequencing (tGBS^®^, Data2Bio technologies, IA, USA) sequences utilizing 10 Ion Proton runs were downloaded. Additional comprehensive information on SNP extraction and quality control was elucidated in Ali et al. (2018) [33]. Briefly, from the original rice diversity panel, which includes 12 parents and 564 introgression lines (ILs), 181 lines and their four parental tGBS^®^ SNPs were extracted. The rice reference genome (Osativa_204_v7.0.fa) was obtained from Phytozome, and raw sequencing reads were preprocessed using Lucy2 [34] to trim low-quality bases. SNPs were extracted using the (Genomic Short-read Nucleotide Alignment Program) GSNAP, filtered based on quality, allele frequency, and heterozygosity, and common SNPs that were found in at least 50% (≈90 EB-SILs) of the combined population were retained for further analysis [35]. Leveraging the physical position of the common SNPs, a physical map was constructed to map QTLs associated with weed competitiveness.

### 2.5. QTL Mapping and Candidate Gene Extraction

The QTL analysis for ESG used the mean values of the phenotypic data gathered, while the relative phenotypic values of morphological traits were determined for each line, and the phenotypic value obtained from the weedy condition divided by the control value of the non-weedy condition was used for the QTL analysis of ESV-related traits. The mapping of QTLs was performed by utilizing single-marker regression analysis using the single-marker analysis (SMA) function in the IciMapping tool v4.1 [36]. For each genetic marker SNP, the phenotypic trait *x* was modeled following the linear equation. In this method, each genetic marker (SNP) is individually analyzed to determine its association with a given phenotypic trait *x*. The relationship between each SNP and trait *x* is modeled with a simple linear equation, where the SNP acts as a predictor of the phenotypic variation in *x*.*y* = *μ* + *αx_SNP_* + *ϵ*
where *y* is the observed weed competitiveness trait, *μ* represents the overall population mean, *α* is the estimated effect of the SNP marker, *x_SNP_* denotes the genotype at marker, and *ϵ* is the residual error term. For each SNP marker, a test statistic *t_m_* was computed to evaluate the association between the marker and the trait, expressed as*t_m_* = *Mean Square^marker^/Mean Square^error^*
where *Mean Square^marker^* represents the variance in *y* explained by the SNP marker, and *Mean Square^error^* is the residual variance. To establish an empirical significance threshold in SMA, permutation testing was applied to control for multiple testing to reduce false positives. Phenotypic values (*y*) were randomly permuted across individuals 1000 times, while marker genotypes (*x_SNP_*) remained fixed, thus generating a null distribution for the test statistic (*t_m_*) under the assumption of no true association. This process preserved the population structure and any confounding effects within the permuted datasets. The permuted *t_m_* values were used to construct an empirical distribution, and the 99th percentile of this distribution was selected as the significance threshold (*t^empirical^*). Markers with original *t_m_* values exceeding *t^empirical^* were identified as significantly associated with the trait. This permutation-based threshold allowed for a more robust association detection by accounting for population structure and reducing false-positive rates in the selected EB-SIL population. The association of a weed competitive trait and a QTL was declared significant once it had a threshold level (−log p(F) ≥ 4.15) based on a 1000 permutation test. To further delimit the confidence interval of each QTL, the 1-LOD drop method from the estimated SNP peek position was followed [37,38]. The gene models lying within the QTL intervals for the discovered QTLs controlling ESG- and ESV-related phenotypes were extracted from the MSU7 Rice Genome Annotation Database (http://rapdb.dna.affrc.go.jp/ accessed on 12 March 2024). Polymorphisms within the candidate genes of the parental lines were obtained from the Rice SNP-Seek Database (https://snp-seek.irri.org/ accessed on 15 March 2024). Further genotyping data on the parents were acquired from the previous Tunable Genotyping-By-Sequencing (tGBS^®^) approach [33]. Subsequently, we identified gene models within the candidate loci that exhibited non-synonymous polymorphism between the parents and were evaluated as the most likely candidates for weed competitiveness.

## 3. Results

### 3.1. ESG Performance of Parental Lines and EB-SILs

The traits involved in ESG are associated with crop competitiveness against weeds (Appendix A). The results showed high phenotypic variation among the lines. Among the 181 EB-SILs and four parents, all ESG-related traits had significant differences (Table 1). Seven out of eight traits had a *p*-value of less than 0.001. The mean ESG performance of the parental lines and top-performing EB-SILs under germination assay are shown in Figure 1. No significant differences were observed among the parental lines and EB-SILs in the second-day germination count, seventh-day germination count, total dry weight, seed vigor index, and coleoptile length. The second-day and seventh-day germination count ranged from 18 to 25 and 23 to 25 in all lines, respectively, while the total dry weight ranged from 0.38 to 0.54 g. The lowest and highest values for coleoptile length were 6.28 and 8.18 cm, respectively, while the lowest and highest values for radicle length were 3.83 and 5.28 cm, respectively. On the other hand, significant differences were observed in the germination rate and radicle length. EB-SILs generally exhibited the highest germination rate, along with the parental lines, Haoannong, ChengHui448, Y134, and WTR1. Additionally, the EB-SILs recorded the highest increase in radicle length among all the lines, ranging from 8.33 to 9.19 cm.

### 3.2. ESV Performance of Parental Lines and EB-SILs

Another associated trait with the weed competitive ability with rice is ESV (Appendix A). High phenotypic variations were observed in all 12 ESV-related traits (Table 2). There were significant differences among the parental lines and EB-SILs with *p*-values of less than 0.001 in all traits except the root dry weight at 28 DAS with a *p*-value of 0.005. On the other hand, all traits had significant differences between the weedy and non-weedy treatments. All ESV-related traits had *p*-values of less than 0.001. A similar high phenotypic variance was also observed in the interaction of lines and conditions. Only a root dry weight at 28 DAS resulted in a *p*-value of 0.008, while the remaining traits have less than 0.001. Weed interference caused reductions in the mean performance of all lines for all ESV-related traits, as shown in the box-plot distribution (Appendix A). For further identification of the effect of weed pressure on the parental lines and EB-SILs, the mean values of phenotypes were plotted in a graph (Figure 2). As early as 14 DAS, all parental lines exhibited tolerance to weeds in terms of plant height. Weed interference did not significantly reduce leaf count at 14 DAS. However, some EB-SILs exhibited extreme phenotypes consistently across all replicates, suggesting a strong vigor response from EB-SILs during early establishment. There were no significant differences in the mean values between the treatments, except for WTR1, which recorded a higher plant height in the weedy condition than in the non-weedy condition. There was also no significant difference between the mean performance of parental lines under non-weedy and weedy conditions at 21 DAS, except for Y134, which had a higher plant height at 21 DAS in the weedy condition. EB-SILs and ChengHui448 had the highest recorded plant height at 21 DAS under weedy conditions. Significant reductions in tiller numbers were recorded in all lines upon subjecting to weedy conditions. However, it was observed that the top-performing EB-SILs had significantly different mean performances in all parameters. The EB-SILs performed the best among all lines across all measured characteristics.

### 3.3. Correlation Analysis Among Measured Traits

A Pearson pairwise comparison was conducted to understand the relationship between ESG-related traits (Figure 3A). The second-day germination count showed a strong positive correlation with the germination rate (r = 0.94, *p* < 0.001) and seed vigor index (r = 0.90, *p* < 0.001). Similarly, the germination rate was positively correlated with the seed vigor index (r = 0.85, *p* < 0.001). The total and average dry weights exhibited a significant positive association (r = 0.59, *p* < 0.001). Conversely, radicle length displayed negative and significant associations with the second-day germination count (r = −0.40, *p* < 0.001), seventh-day germination count (r = −0.29, *p* < 0.001), germination rate (r = −0.37, *p* < 0.001), and seed vigor index (r = −0.37, *p* < 0.001). Furthermore, correlation analysis among all ESV-related traits using relative phenotypic values of non-weedy and weedy treatments was used to assess the response of parental lines and EB-SILs to weed pressure. All traits showed significant positive correlations with each other (Figure 3B). The strongest correlation was observed between the relative seedling vigor index at 28 DAS and relative total dry weight at 28 DAS (r = 1.00, *p* < 0.001). Additionally, a high positive correlation was found between the relative shoot dry weight at 28 DAS and relative total dry weight at 28 DAS (r = 0.98, *p* < 0.001), as well as between the relative root dry weight at 28 DAS and relative total dry weight at 28 DAS (r = 0.86, *p* < 0.001). Conversely, the weakest positive correlation was noted between the relative plant height at 14 DAS, and the relative tiller number at 28 DAS (r = 0.15, *p* < 0.001), and another low positive correlation was found between the relative plant height at 21 DAS and relative leaf count at 28 DAS (r = 0.18, *p* < 0.001).

### 3.4. Principal Component Analysis Among Measured Traits

The Principal Component Analysis (PCA) was conducted to identify the relationship between ESG- and ESV-related traits. For ESG traits, PC1 explains 42.8% and PC2 explains 20.5%, giving a total of 63.3% variance explained by the first two components (Figure 4A). Similarly, for ESV traits, PC1 explains 50.1% and PC2 explains 13.2%, summing to 63.3% variance explained by these two components (Figure 4B). The 15 rice lines that exhibited strong performance were identified by analyzing the main component and loading plot matrix of ESG traits (Appendix A). The lines with excellent ESG features were chosen based on the EB-SILs identified within the region of the seed vigor index and germination rate. These lines showed a favorable correlation between significant ESG traits, such as the second-day germination count, seventh-day germination count, germination rate, and seed vigor index. As measured by ESV-related features, fifteen lines with superior weed competitiveness were identified using principal component and loading plot matrix analysis. These lines have the potential to produce significant yields. Appendix A summarizes the 15 selected best-performing EB-SILs depending upon their location in the biplot matrix (Appendix A). EB-SILs located the nearest with the relative tiller number at 28 DAS were considered the 15 lines with favorable ESV-related traits. The five top-performing EB-SILs (Figure 5), together with their donor and recipient parents, show ESV traits in both weedy and non-weedy environments.

### 3.5. SNP Markers Generated by (tGBS^®^) Sequences for QTL Mapping

A total of 943.4 M raw tGBS^®^ sequencing reads were acquired by extracting the raw tGBS^®^ sequencing reads. These reads were produced using 10 Ion Proton runs from the initial population of 564 (EB-SILs) [33]. The 181 lines used in the current study were selected from a subset of 564 lines taken from combining three populations including the common recipient parent WTR1 and three donor parents Haoannong, ChengHui448, and Y134. To ensure data quality, SNPs with excessive missing data were filtered out using a threshold. Specifically, SNPs with more than 50% (LMD50) missing data were removed by applying a minimum calling rate of 50%. The Low-Missing Dataset was filtered, resulting in the identification of the total number of LMD50 SNPs found in all three sub-populations by analyzing the distinct alignments of every read from the 181 introgression lines in relation to the publicly available reference genome. Sub-population 1, which included 112 lines (110 Introgression Lines and the two parental lines), was denoted as WTR1 X Haoannong, and 4669 LMD50 SNPs were detected. Sub-population 2, which was formed by crossing WTR1 and ChengHui448, had a total of 41 lines. Among these lines, 39 introgression lines (ILs) and the two parental lines were detected, and 5968 LMD50 SNPs were detected. Sub-population 3, also known as WTR1 X Y134, included 28 lines. This included 26 ILs and the two parental lines, and we were able to identify a total of 4435 LMD50 SNPs. By integrating all common LMD50 SNPs across the groups, 3791 LMD polymorphic SNPs were obtained. These 3791 LMD SNPs were unevenly distributed across the genome, and there were 64 large gaps (>1 Mb) across the genome in the distribution of SNPs generated by the tGBS^®^ sequencing as result of plant positive selection, and the monomorphic and non-common SNP markers shared between parents in these regions. Gaps ranging from 6 Mb on chromosome 8 to 8 Mb on chromosome 11 were observed in the generated physical map of the rice genome using the 3791 SNP marker data. Subsequently, these SNPs were used to construct the physical map required for QTL analysis (Figure 6).

### 3.6. Identification of QTLs for Weed Competitive Traits

Nineteen SNPs showed a significant marker–trait association with ESG- and ESV-related traits. QTLs were defined by assuming that closely linked significant markers are in the QTL region (Table 3 and Figure 6). Four of these eight major QTLs (PVE ≥ 10) associated with ESG traits were located on chromosome 2, while two QTLs were on chromosome 12, and one each on chromosomes 3 and 6. The first QTL identified on chromosome 2 was for radicle length, *qRL2*, with a phenotypic variance of 15.25%. On chromosome 2, *qTDWG2*, responsible for the total dry weight during the germination stage of rice (PVE = 13.98%), was also detected. Two QTLs for the seedling vigor index were also found on chromosome 2, where *qSVI2.1* and *qSVI2.2* has a major phenotypic variance of 15.97% and 16.26%, respectively. In addition, another major QTL was detected on chromosome 3, responsible for the seedling vigor index (*qSVI3*), with a phenotypic variance of 11.08%. Another major QTL identified was located on chromosome 6 (*qSVI6*) for early seedling vigor, which has a phenotypic variance of 12.20%. On the other hand, two major QTLs were on chromosome 12: *qGR12* associated with the germination rate (PVE = 10.57%) and *qSVI12* associated with the seedling vigor index (PVE = 13.29%). For ESV-related traits, four QTLs were found on chromosome 10 and one QTL each on chromosomes 1, 3, 4, 5, 8, and 9 (Table 3 and Figure 6). On chromosome 1, *qRPH1* was detected to be related to two ESV-related traits, relative plant height at 14 and 28 DAS, which have a phenotypic variance of 11.29% and 9.94%, respectively. A major QTL was found on chromosome 3 with a phenotypic variance of 14.80% that was associated with the relative tiller number of rice (*qRTN3*), while *qRLC4* identified on chromosome 4 with a phenotypic variance of 11.34% was a QTL for relative leaf count at 28 DAS. Two QTLs linked with relative plant height at 21 DAS were also identified, namely *qRPH5* on chromosome 5 (PVE = 10.10%) and *qRPH*9 on chromosome 9 (PVE = 10.87%). For relative root length at 28 DAS, the major QTL (*qRRL8)* detected was on chromosome 8 (PVE = 10.60%). Moreover, another QTL (*qRRL10)* linked with root length at 28 DAS was on chromosome 10 (PVE = 10.34%), along with *qRTN10* (PVE = 10.24%), *qRLC10.1* (PVE =10.15%), and *qRLC10.2* (PVE = 10.04%), QTLs responsible for a relative tiller number at 28 DAS and relative leaf count at 28 DAS, respectively.

### 3.7. Candidate Genes Associated with ESG and ESV Traits

Among the 18 QTLs identified in this study, there were 480 gene models present within the QTL interval (Appendix A). Out of these, 27 gene models were found to be directly associated with biotic and abiotic stress-tolerant genes, thus considered the most likely candidate genes (Appendix A). These genes were selected for further SNP analysis to find the most promising candidate genes. Fourteen of the most promising candidate gene models were associated with ESG traits, and another thirteen were related to ESV traits. In the most likely candidate gene models, 5286 SNPs were identified between the parents in the Rice SNP-Seek Database. Among those identified SNPs, 24% showed polymorphism between the parents in 27 genes, and most of these polymorphisms (87.5%) were synonymous mutations, with non-synonymous mutations (12.4%) between the parents in 27 genes. Out of the 27 gene models (Appendix A), 19 genes that contain SNPs resulting in non-synonymous mutations were identified as the most promising candidate genes associated with weed competitiveness in rice (Table 4).

## 4. Discussion

In this study, we investigated the genetic basis of weed competitiveness in rice through QTL analysis using a GSR breeding population of 181 EB-SILs derived from the cross between the common recipient parent WTR1 and three donor parents Haoannong, ChengHui448, and Y134. Also, by employing high-density tGBS^®^ SNPs, our study identified nineteen QTL regions associated with weed competitiveness and the specific candidate genes within these regions. Despite weedy pressure, the pronounced phenotypic variation observed among EB-SILs suggests a concrete genetic basis underlying the weed competitiveness trait. This variability indicates a high likelihood of detecting the QTLs governing these traits, underscoring the potential for genetic improvement in rice competitiveness against weeds. Furthermore, the observed phenotypic variations reflect gene segregation upon the backcrossing of genetically distant donor and recipient parents, suggesting that both parental lines contribute favorable traits to the elite EB-SILs [1,2,32]. The EB-SILs exhibited similar mean performances to parental lines across several ESG traits. This parity suggests that desirable ESG-related traits are inherited from both parents and remain expressed in EB-SILs. The seventh-day germination count is of particular significance, which is critical for establishing a competitive advantage over weeds by ensuring timely seedling emergence and higher crop density, thus reducing weed biomass accumulation.

Rapid and uniform seedling emergence, measured by the seed vigor index, is crucial for rice competitiveness. Coleoptile and radicle length are also essential for resource acquisition and crop competitiveness. Seed vigor, indicating rapid and uniform germination, is pivotal for plant establishment [39]. The second-day germination count and germination rate positively correlated with the seed vigor index, supporting their importance for rice competitiveness against weeds. These associations align with previous findings highlighting the importance of these ESG traits in determining rice competitiveness against weeds. However, inconsistencies arise regarding radicle length correlations with ESG traits, contrasting with previous reports [40]. A fundamental principle of crop-weed competition underscores the advantage of early establishment, emphasizing the critical role of the emergence time in field competitiveness [25]. Accordingly, the second-day germination count, seventh-day germination count, germination rate, and seed vigor index emerge as pivotal ESG traits influencing rice competitiveness against weeds.

The mean performances of EB-SILs and parental lines in ESV traits were notably diminished under weedy conditions. Significant differences were observed in plant height, leaf count, and tiller number at 28 DAS among all lines. Conversely, no significant differences were noted in root length, shoot dry weight, root dry weight, total dry weight, and seedling vigor at 28 DAS under non-weedy conditions, consistent even in weedy conditions. This reduction in EB-SIL performance in weedy conditions can be attributed to the competitive pressure exerted by jungle rice, leading to interspecific competition and, subsequently, lower rice performance compared to non-weedy conditions. This observation aligns with previous studies reporting decreased values for plant height, tillering ability, and chlorophyll content under weedy conditions [27]. A decrease in rice tillers was evident with increasing weed density. In contrast, a negative correlation between *Echinochloa* spp., weed dry weight, and rice root dry weight suggested a direct impact of weeds on root development [17,41]. Similarly, a decrease in the seedling vigor index correlated with higher weed density, implying the detrimental effects of weed competition on rice seedling vigor. These findings underscore the negative impacts of weeds on rice growth and development, emphasizing the importance of effective weed management strategies and adoption of weed competitive varieties [41]. Notably, no significant differences in plant height between weedy and non-weedy treatments for parental lines suggest genetic stability under both conditions, indicating the resilience of genotypes to weed competition.

All ESV-related traits exhibited significant and positive correlations with each other in this study. These ESV-related traits were also significantly and positively correlated in the study conducted by Dimaano et al. (2017) [27]. A positive correlation between plant height and dried vegetative crop biomass under non-weedy and weedy conditions was also observed by [42]. Additionally, dried vegetative crop biomass was closely related to tiller number, vigor ratings, and canopy ground cover [19]. The results observed in the study, as well as in other studies, revealed that the ability of the seedlings to emerge and have vigorous growth, also known as seedling vigor, is governed by different ESV-related traits, where it is the sum of various properties of a plant associated with the rate and uniformity of seedling growth [39]. Weed tolerance pertains to the ability of the crop to have a high yield despite weeds in the field. In this study, the tiller number at 28 DAS was used to associate with the grain-yielding capacity of rice plants, as the tiller number is positively correlated with grain yield [27]. During the vegetative growth stage, the tillering number is a trait highly associated with the panicle number, which is a critical yield component of rice [43]. Identifying top-performing EB-SILs based on the seed vigor index, germination rate, and tiller number at 28 DAS provides insights into selecting lines with desirable ESG and ESV traits. The top-ranking lines exhibit promising correlations among key ESG traits and between ESV-related traits and the tiller number at 28 DAS, suggesting their potential for high yield and weed competitiveness.

Weed competitiveness is an untargeted trait that was not considered during the population development of EB-SILs by assuming no correlation exists between weed competitiveness and the selected traits involved in population development, yield under different conditions like irrigated, rainfed, drought, salinity, submergence, and low input conditions [33]. The selected EB-SIL populations could be roughly considered as random segregation populations for mapping QTLs associated with weed completeness. Previously, similar approaches were employed in elite breeding lines derived from targeted breeding programs to identify favorable QTLs and simultaneously enhance the elite lines for various biotic and abiotic stresses [1,2,37]. In the development of an EB-SIL rice population under selection pressure, linkage-based interval mapping proved less successful due to segregation distortion and the elimination of rice lines with unfavorable recombination through selection, which impeded the precise identification of QTLs [44]. Artificial selection pressure in a breeding program favored specific alleles, reducing unfavorable linkages and the occurrence of necessary recombination events for interval mapping. Also, the distorted segregation patterns in a positively selected breeding population violated the assumptions required for precise linkage analysis [45]. In contrast, SMA was more applicable in such cases as it directly tested the association between individual markers and traits without relying on recombination information [44]. SMA’s independence from recombination and its simpler statistical model made it more reliable for detecting marker–trait associations under conditions of selection pressure, where linkage-based methods struggled due to altered genetic diversity and fixed genomic regions. In the current weed competitiveness screening, the population was advanced using the backcross breeding approach, retrieving only a small number of genomic introgression fragments from a donor parent. This limited the number of genomic introgression fragments present in the EB-SILs. However, tunable genotyping-by-sequencing (tGBS^®^) for genotyping yielded a substantial number of polymorphic markers, enabling a clear distinction of genomic introgression fragments [33,46]. These markers have the potential to provide a comprehensive understanding of genetic variation in the population.

QTL mapping for weed competitiveness involved using 3791 LMD SNP markers, resulting in the mapping of 19 significant SNPs linked to QTLs associated with this trait through marker–trait association. To assess the novelty of our findings, the putative QTL regions identified for weed competitiveness traits were compared with previously reported QTLs. This comparison was based on the physical positions of the associated markers in the Nipponbare genome, using information from the International Rice Genome Sequencing Project (http://rgp.dna.affrc.go.jp/IRGSP/, accessed on 24 May 2024). In ESG-related traits, a total of eight QTLs were identified. Four QTLs were detected on chromosome 2, linked with the radicle length (*qRL*2), total dry weight (*qTDWG*2), and seed vigor index (*qSVI*2.1 and *qSVI*2.2) of the rice plants. Two were identified on chromosome 12, governing the germination rate (*qGR*12) and seed vigor index (*qSVI*12). Two other QTLs related to the seed vigor index were found on chromosome 3 (*qSVI*3) and 6 (*qSVI*6). The QTLs detected for the ESG-related traits in this study differed from those reported QTLs associated with the said traits [47,48]. There were 11 QTLs associated with ESV-related traits. On chromosome 1, a QTL (*qRPH*1) was detected that was linked with the relative plant height trait at 14 DAS and relative plant height at 28 DAS. Then, four QTLs were located on chromosome 10, which were associated with relative leaf count at 28 DAS (*qRLC*10.1 and *qRLC*10.2), relative tiller number at 28 DAS (*qRTN*10), and relative root length at 28 DAS (qRRL10), which was consistent with previous studies [47,48,49]. Moreover, there were also QTLs detected for relative plant height at 21 DAS on chromosomes 5 and 9 (*qRPH*5 and *qRPH*9), relative leaf count at 28 DAS on chromosome 4 (*qRLC*4), relative tiller number at 28 DAS on chromosome 3 (*qRTN*3), and relative root length at 28 DAS on chromosome 8 (*qRRL*8). The detected QTL was linked with plant height at 14 and 28 DAS (*qRPH*1) and co-localized with qPH-14.1, identified for rice plant height at 14 DAS by [50]. However, other QTLs found on chromosome 1 in the study of Dimaano et al. (2020) [50], which are associated with plant height at 21 and 28 DAS, were not detected in the current study. Instead, novel QTLs for relative plant height at 21 DAS were determined on chromosomes 5 and 9, *qRPH*5 and *qRPH*9, respectively. Moreover, the rest of the identified ESV trait-related QTLs were all novel.

Among the 18 QTLs identified in this study, there were 480 gene models present within the QTL interval, and we further focused on gene models within the QTL intervals associated with biotic and abiotic stress tolerance, prioritizing those containing non-synonymous SNPs, as stress tolerance is directly and biologically relevant to weed competitiveness. The genes involved in stress tolerance contribute to key adaptive traits—such as resilience to drought, nutrient scarcity, and pest pressures—that align closely with competitive advantages in plants facing weed interference. By enabling plants to survive and thrive under environmental stressors, these genes inherently enhance a plant’s ability to compete for resources in field conditions. Non-synonymous SNPs, which induce amino acid changes, were specifically prioritized for their potential impact on protein function, which can lead to meaningful phenotypic adaptations associated with both stress resilience and competitiveness. Additionally, due to the high cost and complexity of expression data for large gene sets, this approach strategically narrows the focus to biologically plausible candidate genes most likely to influence weed competitiveness, without diluting the analysis with less relevant genes. While other functionally annotated genes might hypothetically contribute, stress-tolerance genes with functional SNPs present the most likely targets for uncovering impactful genetic variation in competitive traits. To enhance the precision of our QTL analysis, we utilized a whole-genome sequencing strategy for the parental lines. This approach aimed to narrow down the candidate genes within the QTL intervals. Since QTLs mapped in bi-parental populations are confined to the loci present in the gene pool of the founder parents, we focused on analyzing non-synonymous mutations between the parental sequences within the QTL interval [37,38] (Appendix A and Table 4).

Overall, the whole-genome sequence of parents and gene expression analysis proved to be an effective strategy to narrow down the candidate genes in the QTL intervals. Among these, *ASR*4 (abscisic acid-stress-ripening-inducible4 protein), encoded by *LOC_Os01g73250,* is associated with the abscisic acid (ABA) response, which can regulate seed dormancy and germination. ABA influences weed competitiveness by affecting seed germination timing, potentially giving rice seedlings an advantage over weeds [51]. Among these genes, *LOC_Os09g24560* and *LOC_Os02g15340*, both encoding putative no apical meristem proteins and belonging to the *NAC* family of plant transcription factors, are implicated in meristem regulation, potentially influencing weed competition through growth modulation [52]. Additionally, *LOC_Os09g24800*, a putative MYB family transcription factor, *and LOC_Os02g15350*, encoding a dof zinc finger domain-containing protein, underscore the regulatory mechanisms involved in response to weed pressure [53]. The presence of genes like *LOC_Os10g33940*, encoding an auxin response factor 18 (*ARF22*), and *LOC_Os10g33960*, expressing a START domain-containing protein (*OSHB2*), highlights the involvement of hormonal signaling pathways and transcriptional regulation in weed competitiveness [54,55]. Furthermore, genes such as *LOC_Os10g34020* and *LOC_Os10g34430*, encoding putative glutathione S-transferase and Dicer proteins, respectively, suggest the importance of stress response mechanisms in weed competition [56]. Notably, genes like *LOC_Os12g10720* and *LOC_Os12g10730*, both encoding glutathione S-transferases, and *LOC_Os12g12580*, encoding an NADP-dependent oxidoreductase, indicate the role of detoxification processes in enhancing rice competitiveness against weeds [57]. Moreover, genes like *LOC_Os02g50240*, encoding glutamine synthetase 1;1, *LOC_Os02g50330*, encoding an RNA-dependent RNA polymerase, *LOC_Os06g04070*, encoding an arginine decarboxylase, and *LOC_Os06g04200*, encoding a starch synthase, further contribute to the multifaceted response of rice plants to weed competition, emphasizing the intricate interplay between various molecular pathways in shaping weed competitiveness in rice [58]. The ZF-HD protein encoded by *LOC_Os09g24820* is a transcription factor that regulates multiple developmental processes in rice plants. While its specific role in weed competitiveness is not well understood, transcription factors like ZF-HD are often associated with stress responses and growth regulation. It is plausible that ZF-HD may indirectly influence weed competitiveness by modulating the expression of genes involved in stress tolerance or developmental pathways that affect plant vigor. ZF-HD might regulate the expression of genes involved in root architecture or nutrient uptake, traits crucial for outcompeting weeds [52]. On the other hand, *LEA15* (Late embryogenesis abundant protein 15), encoded by *LOC_Os02g15250,* is a protein known for protecting plants from various stresses, including drought and salinity. Early seedling vigor and germination in rice are closely linked to stress tolerance during the critical early stages of growth. LEA proteins like *LEA15* are involved in maintaining cellular hydration and stabilizing proteins and membranes under stress conditions, which can contribute to improved seedling vigor and germination rates. Enhancing stress tolerance during germination and early seedling growth, *LEA15* may indirectly enhance competitiveness against weeds by ensuring a robust start for rice plants [59,60]. Interestingly, ZF-HD and LEA proteins are part of the complex regulatory network governing plant responses to environmental stimuli. While ZF-HD proteins may be more directly involved in transcriptional regulation, LEA proteins act as molecular chaperones, safeguarding cellular components during stress. Despite their distinct roles, both types of proteins ultimately contribute to the overall fitness and competitiveness of rice plants by ensuring proper growth and development, especially during the early stages when plants are most vulnerable to weed competition and environmental stresses [52,59,60].

The study of genetic traits related to weed competitiveness in rice is still developing, with limited QTLs and candidate gene information available. While some QTLs have been identified for traits such as early vigor, root architecture, and allelopathy, the overall number remains modest. The QTLs and candidate genes identified in our study have significant implications for sustainable agriculture, as improving weed competitiveness can reduce herbicide reliance, promote ecological balance, and enhance yield stability in DSR farming systems. Our findings deepen the understanding of early ESG and ESV, supporting marker-assisted and genomic selection strategies in breeding programs. Moreover, our research findings will provide a foundation for employing advanced biotechnological approaches, such as gene editing and cloning, to manage weeds like jungle rice. Collectively, these efforts will play a critical role in developing resilient rice varieties that can enhance agricultural productivity while ensuring environmental sustainability.

## 5. Conclusions

Our work investigated the genetic factors that contribute to the ability of rice plants to compete with weeds. We used QTL analysis on a mapping population of 181 EB-SILs created by crossing the recipient parent WTR1 with three different donor parents. We used high-density tGBS^®^ SNPs to detect a total of 19 SNPs, which showed significant marker–trait associations, leading to the identification of 18 major QTLs. Key QTLs for ESG traits were located on chromosomes 2, 3, 6, and 12, while ESV-related QTLs were found on chromosomes 1, 3, 4, 5, 8, 9, and 10. Hence, this study detected 17 novel QTLs that could aid in breeding rice with improved weed competitive ability during its early vegetative stage. Four key QTLs on chromosome 2 were associated with radicle length (*qRL*2), total dry weight (*qTDWG*2), and seedling vigor index (*qSVI*2.1 and *qSVI*2.2), making it a critical region for early seedling growth. Chromosome 10 was identified as a key region for ESV-related traits, with four major QTLs linked to root length, tiller number, and leaf count at 28 DAS, emphasizing its significant role in enhancing weed competitiveness in rice. Among the 480 gene models within the identified QTL regions, 27 were associated with stress tolerance. Of these, 19 genes contained SNPs with non-synonymous mutations, making them strong candidates for enhancing weed competitiveness in rice. Our analysis also highlights the negative impacts of weeds on rice growth and development, emphasizing the importance of effective weed management strategies. The identified QTLs and candidate genes provide valuable insights into the genetic mechanisms underlying weed competitiveness in rice, with implications for sustainable agriculture. Enhancing weed competitiveness in rice can reduce herbicide usage, increase yield stability, and promote resource use efficiency, contributing to the economic and environmental sustainability of the rice production system. Overall, these findings would assist rice breeders in developing rice varieties with a competitive advantage against weeds suitable for DSR systems. In addition, the candidate genes associated with ESG and ESV traits could be further studied to understand better the expression of these genes under high weed pressure.

## Figures and Tables

**Figure 1 biology-14-00413-f001:**
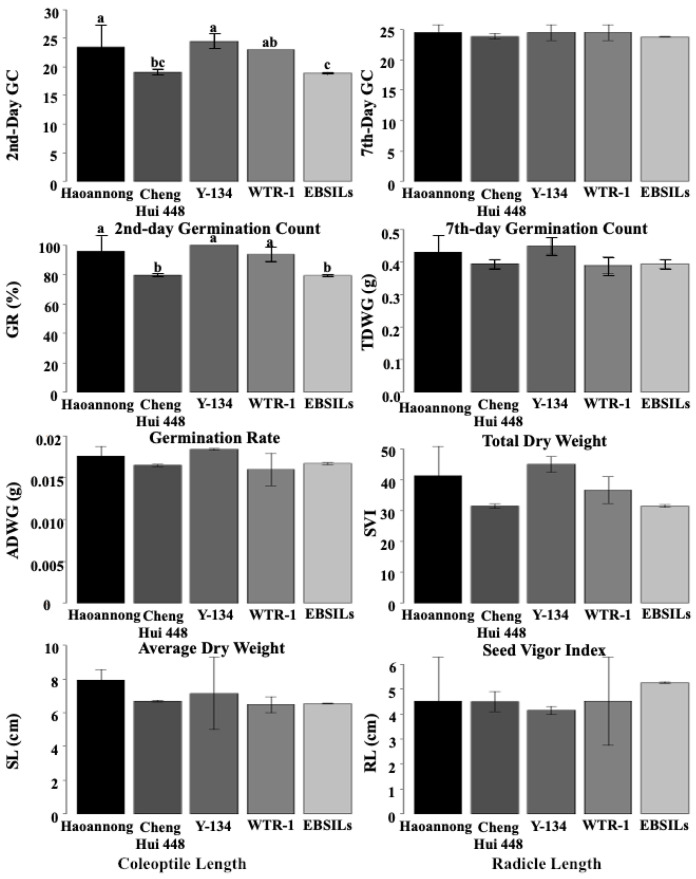
Mean performance of parental lines and Early Backcross Selective-Introgression Lines (EB-SILs) was assessed under germination assay for early seed germination (ESG) traits. The vertical bars in the graph represent the mean phenotypic performance of these lines. Traits with no letter above the bars indicate no significant difference among the lines, while the presence of letters above the bars indicates significant differences among the lines. Different letters denote significantly different means.

**Figure 2 biology-14-00413-f002:**
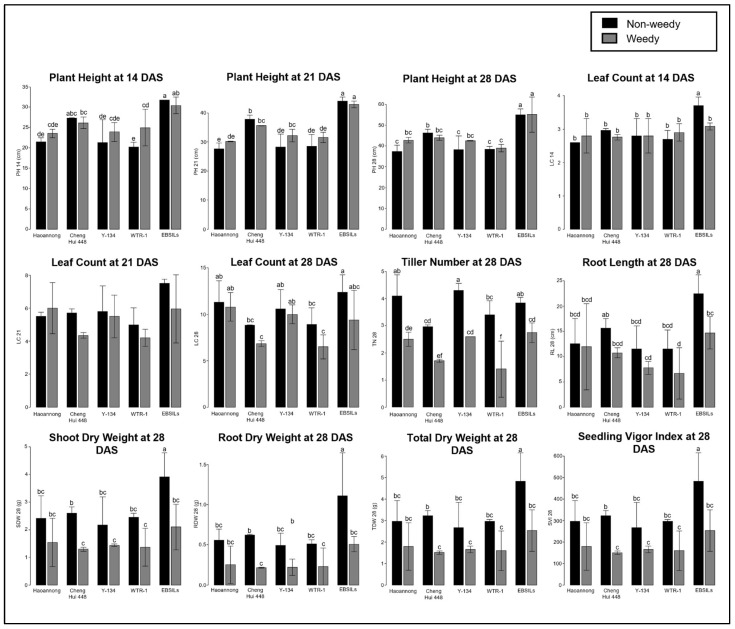
Mean performance of parental lines and Early Backcross Selective-Introgression Lines (EB-SILs) was assessed under both non-weedy and weedy conditions for early seedling vigor (ESV). The vertical bars in the graph represent the mean phenotypic performance of these lines. Traits with no letter above the bars indicate no significant difference among the lines, while letters above the bars indicate significant differences among the lines. Different letters denote significantly different means.

**Figure 3 biology-14-00413-f003:**
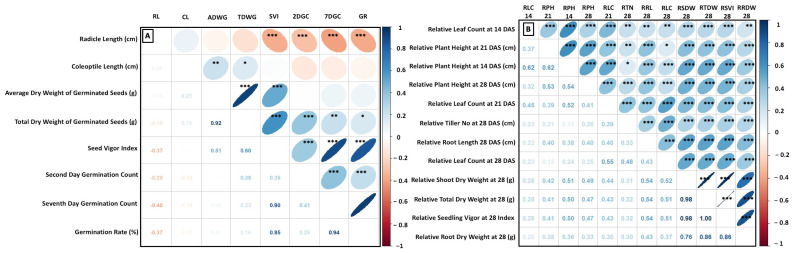
The heat map displays the correlation coefficients for traits related to weed competitiveness. Heat map (**A**) shows the correlation coefficients for early seed germination (ESG) traits, while heat map (**B**) illustrates the correlation coefficients for early seedling vigor (ESV) traits. Significant * 0.05 ≥ *p*-value ≥ 0.01; ** 0.01 ≥ *p*-value ≥ 0.001; *** *p*-value ≤ 0.001.

**Figure 4 biology-14-00413-f004:**
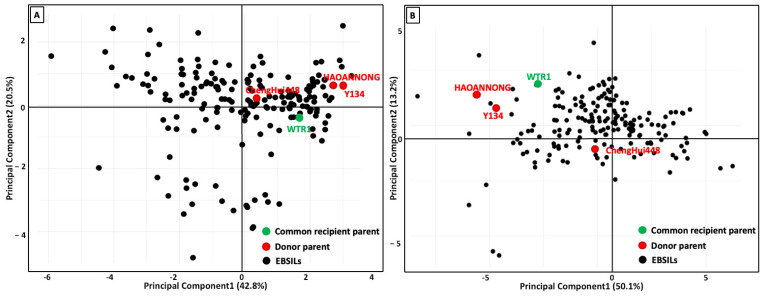
Principal Component Analysis (PCA) was conducted to determine traits related to rice competitiveness against weeds. Plot (**A**) shows the score plot for the first two principal component (PC) scores, PC1 versus PC2, depicting the measurements for early seed germination (ESG) traits across all genotypes. Plot (**B**) displays the score plot for PCA of early seedling vigor (ESV) traits, again showing PC1 versus PC2 for all genotypes.

**Figure 5 biology-14-00413-f005:**
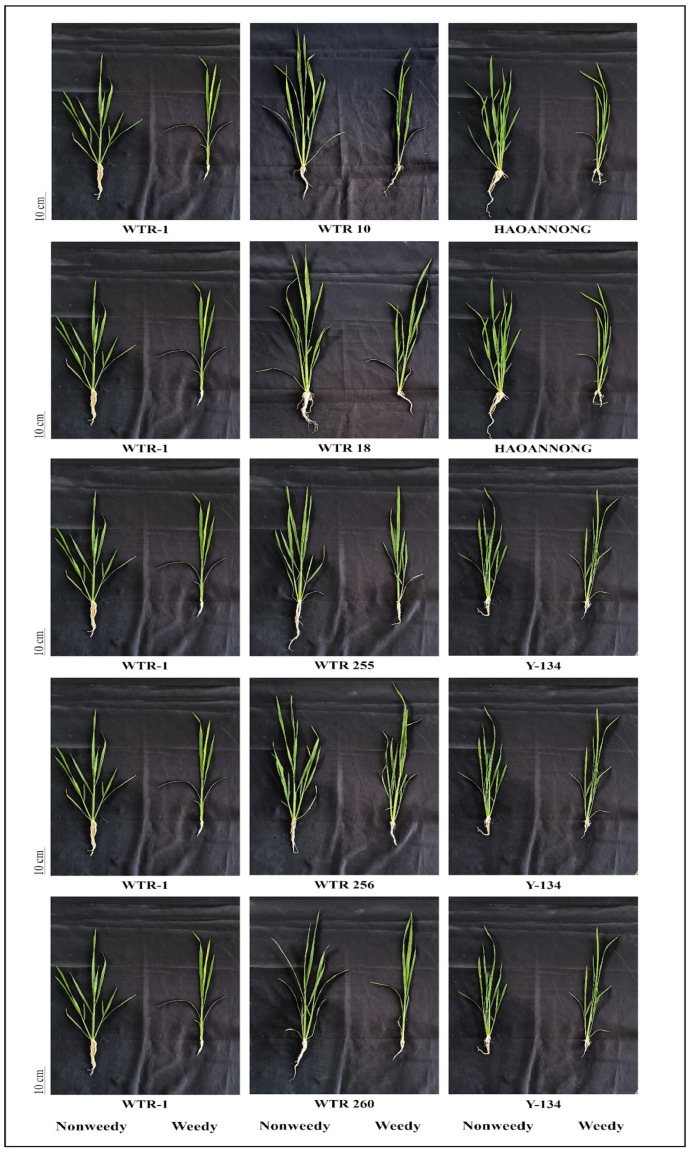
Five selected early backcross selective-introgression lines (EB-SILs) along with their recipient and donor parents showing early seedling vigor (ESV) traits—abbreviations: NW, non-weedy condition; W, weedy condition.

**Figure 6 biology-14-00413-f006:**
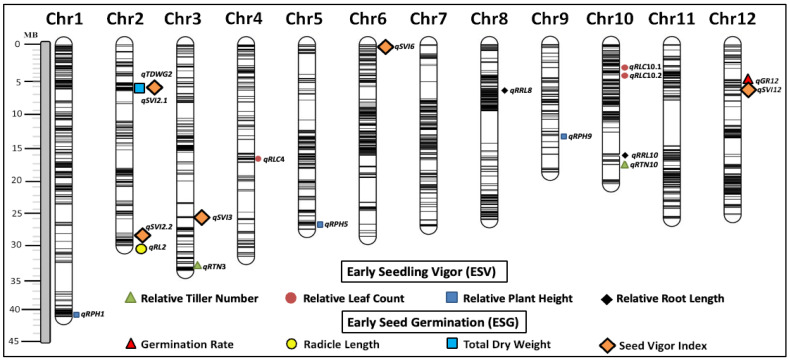
Chromosomal distribution of Low Missing Data 50 (LMD50) single nucleotide polymorphisms (SNPs) and identified quantitative trait loci (QTLs) associated with early seed germination (ESG) and early seedling vigor (ESV) traits. QTLs were located on the chromosome based on the physical position of the SNP marker.

**Table 1 biology-14-00413-t001:** Descriptive statistics of early seed germination (ESG) traits.

Traits	Min.	Max.	Mean	Sum of Squares	Mean Square	F-Value	*p*-Value
Second-Day Germination Count	3	25	19.01	5671	30.821	7.082	0.0000 ***
Seventh-Day Germination Count	19	25	23.77	356.9	1.940	1.553	0.00147 **
Germination Rate (%)	15	100	79.77	88,891	483.1	8.34	0.0000 ***
Coleoptile Length (cm)	3.16	9.79	6.536	876.7	4.765	8.927	0.0000 ***
Radicle Length (cm)	1.12	15.4	5.24	3274	17.80	5.614	0.0000 ***
Total Dry Weight of Germinated Seeds (g)	0.13	0.69	0.3953	0.6862	0.0037	2.145	0.0000 ***
Average Dry Weight of Germinated Seeds (g)	0.0054	0.03	0.0166	0.0010	5.597 × 10^−6^	2.032	0.0000 ***
Seed Vigor Index	5.25	63	31.68	21560	117.2	4.986	0.0000 ***

Significant codes: ** 0.01 ≥ *p*-value ≥ 0.001; *** *p*-value ≤0.001.

**Table 2 biology-14-00413-t002:** Descriptive statistics of early seedling vigor (ESV) traits.

Traits	Non-Weedy			Weedy			ANOVA Result		
	Min.	Max.	Mean	Min.	Max.	Mean	G	T	G*T
Plant Height at 14 DAS (cm)	13.2	41.2	25.89	13	36.5	24.79	***	***	***
Plant Height at 21 DAS (cm)	22.2	54.9	36.68	17.2	51.4	34.1	***	***	***
Plant Height at 28 DAS (cm)	23	65.6	45.79	20	90	42.88	***	***	***
Leaf Count at 14 DAS	2	5	3	1	5	2.86	***	***	***
Leaf Count at 21 DAS	2	10	5.87	2	8	4.44	***	***	***
Leaf Count at 28 DAS	3	17	9.5	3	16	6.56	***	***	***
Number of Tiller at 28 das	1	6	2.99	1	5	1.71	***	***	***
Seedling Vigor Index	141	655	352.37	26	419	149.44	***	***	***
Shoot Dry Weight (g)	1.2	4.87	2.85	0.13	3.15	1.29	***	***	***
Root Dry Weight (g)	0.06	1.68	0.67	0.01	1.04	0.21	**	***	**
Total Dry Weight (g)	1.41	6.55	3.52	0.26	4.19	1.5	***	***	***
Root Length (cm)	4.5	37	16.33	2.8	24.9	9.64	***	***	***

Significant codes: ** 0.01 ≥ *p*-value ≥ 0.001; *** *p*-value ≤ 0.001. Abbreviations: G, genotype; T, treatment; G*T, genotype × treatment interaction.

**Table 3 biology-14-00413-t003:** Quantitative trait loci (QTLs) associated with weed competitiveness at early germination and early vegetative stages in early-backcross selective introgression lines (EB-SILs) breeding population by marker trait association analysis (SMA).

No.	QTL ^a^	Trait	Chr.	Position ^b^	Associated Marker ^c^	LOD ^d^	PVE ^e^	Additive Effect ^f^	Tolerance Allele ^g^
1	*qGR12*	Germination Rate	12	5720211–6950257	S12_6548722	4.51	10.57	−6.15	Haoannong
2	*qRL2*	Radicle Length	2	34379631–34576493	S2_34576493	6.68	15.25	0.55	ChengHui448
3	*qTDWG2*	Total Dry Weight	2	8474520–8752801	S2_8699045	6.08	13.98	−0.02	Y134
4	*qSVI2.1*	Seed Vigor Index	2	8474495–8752801	S2_8474495	7.03	15.97	−4.04	Y134
5	*qSVI2.2*		2	30478421–30791659	S2_30791659	7.17	16.26	−3.62	Y134
6	*qSVI3*		3	25401607–25452773	S3_25401672	4.74	11.08	−3.25	Y134
7	*qSVI6*		6	1542513–1877725	S6_1698496	5.26	12.2	−2.81	Y134
8	*qSVI12*		12	6950207–7011126	S12_6950257	5.76	13.29	−3.45	Y134
9	*qRPH1*	Relative Plant Height at 14 DAS	1	42171596–42617013	S1_42549502	4.84	11.29	−4.27	WTR1
10	*qRPH*5	Relative Plant Height at 21 DAS	5	28497478–28567356	S5_28525048	4.29	10.10	−4.61	Y134
11	*qRPH9*		9	14626500–14826499	S9_14725794	4.64	10.87	−5.19	Y134
12	*qRPH1*	Relative Plant Height at 28 DAS	1	42171596–42617013	S1_42171596	4.23	9.94	−3.47	Y134
13	*qRLC10.1*	Relative Leaf Count at 28 DAS	10	416500–516499	S10_466091	4.32	10.15	−4.32	ChengHui448
14	*qRLC10.2*		10	1393500–1493499	S10_1441265	4.27	10.04	−4.61	ChengHui448
15	*qRLC*4		4	1753300–1753386	S4_1753338	4.86	11.34	4.42	ChengHui448
16	*qRTN*3	Relative Tiller Number at 28 DAS	3	35948030–35965394	S3_35948030	6.47	14.80	−6.18	Haoannong
17	*qRTN10*		10	18051000–18164000	S10_18105284	4.36	10.24	−5.46	Haoannong
18	*qRRL8*	Relative Root Length at 28 DAS	8	7492455–7777214	S8_7541070	4.52	10.60	−4.42	Haoannong
19	*qRRL10*		10	18060329–18452140	S10_18228061	4.41	10.34	−4.88	Haoannong

^a^ Closely linked markers are assumed as the same QTL, ^b^ Physical position of markers on chromosomes, ^c^ Marker associated with QTL. ^d^ Logarithm of odds, ^e^ Proportion of phenotypic variance explained. ^f^ Positive/negative values indicate that additive effect that can increase trait values. ^g^ Tolerance allele provided by parental line.

**Table 4 biology-14-00413-t004:** Total number of genes within the candidate region containing non-synonymous SNPs. These genes were proposed as the most promising candidates for traits related to weed competitiveness.

S.No	Stage	Locus	Annotation	SNPs in 3K RGP	Haoannong		ChengHui448		Y134		WTR-1	
					P-SNPs	NS-SNPs	P-SNPs	NS-SNPs	P-SNPs	NS-SNPs	P-SNPs	NS-SNPs
1	Early Seedling Vigor	LOC_Os01g73250	Abscisic stress-ripening, putative, expressed (*ASR4*)	146	5	1	10	2	23	5	8	2
2		LOC_Os09g24560	No apical meristem protein, putative, expressed	39	0	0	5	1	3	1	5	1
3		LOC_Os09g24800	MYB family transcription factor, putative, expressed	52	1	1	2	1	2	1	2	1
4		LOC_Os09g24820	ZF-HD protein dimerization region containing protein expressed	267	2	0	27	1	34	3	37	3
5		LOC_Os10g33940	Auxin response factor 22, putative, expressed (*ARF22*)	251	27	4	26	4	23	4	23	4
6		LOC_Os10g33960	START domain-containing protein expressed (*OSHB2*)	325	20	1	24	2	25	2	25	2
7		LOC_Os10g34020	Glutathione S-transferase, putative, expressed (*GSTU47*)	354	24	0	13	2	18	1	31	1
8		LOC_Os10g34430	Dicer, putative, expressed (*DCL3B*)	469	21	8	22	8	26	9	26	9
9	Early Seed Germination	LOC_Os12g10720	Glutathione S-transferase, putative, expressed (*GSTZ1*)	258	2	0	7	0	0	0	11	1
10		LOC_Os12g10730	Glutathione S-transferase, putative, expressed (*GSTZ2*)	199	0	0	6	1	0	0	7	1
11		LOC_Os12g12580	NADP-dependent oxidoreductase, putative, expressed (*CLPC2*)	306	18	1	65	11	6	0	69	13
12		LOC_Os02g15250	Late embryogenesis abundant domain-containing protein, putative, expressed (*LEA15*)	88	0	0	1	1	11	6	10	5
13		LOC_Os02g15340	No apical meristem protein, putative, expressed	113	0	0	0	0	12	5	12	4
14		LOC_Os02g15350	dof zinc finger domain-containing protein, putative, expressed (*RPBF*)	267	3	1	3	0	14	3	52	3
15		LOC_Os02g50240	Glutamine synthetase, a catalytic domain-containing protein, expressed (*GLN1;1*)	194	2	0	8	1	10	1	10	1
16		LOC_Os02g50330	RNA-dependent RNA polymerase, putative, expressed (*RDR1*)	344	6	0	17	1	18	1	6	0
17		LOC_Os06g04070	pyridoxal-dependent decarboxylase protein, putative, expressed (*ACD1*)	343	2	1	0	0	6	3	0	0
18		LOC_Os06g04200	Starch synthase, putative, expressed (*WX1*)	403	6	0	0	0	28	1	2	0

3k RGB (3000 Rice Genomes Project-https://snp-seek.irri.org, accessed on 20 May 2024), P-SNPs (Polymorphic Single Nucleotide Polymorphisms), NS-SNPs (non-synonymous Polymorphic Single Nucleotide Polymorphisms).

## Data Availability

This research article offers extensive data supporting its conclusions, presented in figures, tables, and additional Appendix A.

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
