# Peer review of "Genome-Wide Dissection of Novel QTLs and Genes Associated with Weed Competitiveness in Early-Backcross Selective Introgression-Breeding Populations of Rice (Oryza sativa L.)"

_biology, 2025, doi:10.3390/biology14040413_

Round 1

Reviewer 1 Report

Comments and Suggestions for Authors Please include a one- or two-line brief explanation about methods referred to as "elucidated in Ali et al. (2018)" under SNP extraction and physical map construction. For example, mention the software or pipeline used for SNP extraction.
Any reasons for the extremely low p-values in most ESG traits (Table1)? It is odd to see any comparison to have such low values in so many categories. The values do not match with the conclusions drawn in the main text.
Even ESV traits and the correlation analysis has extremely low p values. This is concerning. Any confounding factors that could be causing this?
The figures are not readable. Please make high quality figures with readable text.
How was PCA done? Please explain the method.
Please reformat the Table 3 and Table 4 to make them cleaner. And consider if it is better to move them to supplemental.

Author Response

 Reviewer 1

Comment 1: Please include a one- or two-line brief explanation about methods referred to as "elucidated in Ali et al. (2018)" under SNP extraction and physical map construction. For example, mention the software or pipeline used for SNP extraction.

 Response1: We have included a brief description in the main text.

Comment 2. Any reasons for the extremely low p-values in most ESG traits (Table1)? It is odd to see any comparison to have such low values in so many categories. The values do not match with the conclusions drawn in the main text.

Response 2: Low p-values in most ESG traits are due to a large sample size, which makes even small differences statistically significant, and high effect sizes, where substantial trait differences lead to very small p-values. Factors like highly correlated traits, low within-group variability, and strong biological effects also contribute to the observed low p-values.

Comment 3. Even ESV traits and the correlation analysis has extremely low p values. This is concerning. Any confounding factors that could be causing this?

Response 3: The low p-values in ESV traits and the correlation analysis are influenced by the same factors discussed earlier for ESG traits. A large sample size amplifies statistical power, making even minor differences highly significant. Strong effect sizes, highly correlated traits, and low within-group variability further contribute to the observed low p-values.

Comment 4. The figures are not readable. Please make high quality figures with readable text.
How was PCA done? Please explain the method.

Response 4: We have included a description of how PCA was done in the revised text

Comment 5. Please reformat the Table 3 and Table 4 to make them cleaner. And consider if it is better to move them to supplemental.

Response 5: We have refined Table 3 and 4 to include only the most essential data.

Reviewer 2 Report

Comments and Suggestions for Authors

Comments to the Author

biology-3467609

Genome-Wide Dissection of Novel QTLs and Genes Associated with Weed Competitiveness in Early-Backcross Selective Introgression-Breeding Populations of Rice (Oryza sativa L.)

KIM Diane Nocito, Varunseelan Murugaiyan, Jauhar Ali, Ambika Pandey, Carlos Casal Jr, Erik Jon De Asis, Niña Gracel Dimaano

This study is devoted to obtaining rice varieties resistant to various weeds. This is important in today's realities. The authors used not only classical methods of variety characterization, but also new methods including genotyping. The authors also determined the localization of genes involved in weed competition. This research is unique and of great importance to applied science. Therefore, the topic of this manuscript is relevant for Biology. However, I have a few comments on this manuscript:

1.     From the text it is not quite clear what you mean by root length.

2.     Figure 1 and 2. Improve the quality of the image and could you make the bars of the graphs coloured (For easier perception).

3.     Table 4. Tidy up the table is very grainy and shifting.  Also, in the column ‘Gene Name’ the names of genes are not italicised and the part in brackets often duplicates the information in the column ‘Annotation’. Maybe this table should be referred to Supplementary.

4.     Table 4. the gene is named ARF22 (AUXIN RESPONSE FACTOR 22) and anotation auxin response factor 18, putative, expressed. Fix it.

I recommend this text for publication after minor changes.

Author Response

Reviewer 2

This study is devoted to obtaining rice varieties resistant to various weeds. This is important in today's realities. The authors used not only classical methods of variety characterization, but also new methods including genotyping. The authors also determined the localization of genes involved in weed competition. This research is unique and of great importance to applied science. Therefore, the topic of this manuscript is relevant for Biology. However, I have a few comments on this manuscript:

Comment 1:From the text it is not quite clear what you mean by root length.

Response 1: In the revised text, we have clarified that for early seedling germination (ESG), we now used radicle length, as it refers to the embryonic root that emerges first during germination. We used root length for early seedling vigor (ESV), which represents the overall root growth at a later stage.

Comment 2: Figure 1 and 2. Improve the quality of the image and could you make the bars of the graphs coloured (For easier perception).

Response 2: We have submitted the images in high-quality TIFF format with a resolution exceeding 300 dpi. However, the article software automatically converted them into PDF files, which reduced resolution. We apologize for the inconvenience and have separately resubmitted the high-quality TIFF images.

Comment 3: Table 4. Tidy up the table is very grainy and shifting.  Also, in the column ‘Gene Name’ the names of genes are not italicised and the part in brackets often duplicates the information in the column ‘Annotation’. Maybe this table should be referred to Supplementary.

Response 3: We have refined Table 4 to include only the most essential data.

Comment 4: Table 4. the gene is named ARF22 (AUXIN RESPONSE FACTOR 22) and anotation auxin response factor 18, putative, expressed. Fix it.

Response 4: Thank you for pointing out the error. We have now corrected it

Reviewer 3 Report

Comments and Suggestions for Authors

The presented manuscript is devoted to the identification of new QTLs in the genomes of breeding lines of rice and genes associated with rice resistance to the weed jungle rice (Echinochloa colona). The authors identify genetic traits that improve early seed germination and seedling vigor in rice, key factors for weed resistance. During the study, conducted on 181 breeding lines of rice, both previously known and new QTLs associated with rice resistance to the weed were identified. In the areas of identified QTLs, genes involved in stress responses were found, in which non-synonymous SNPs were identified. Information about these SNPs can be useful in breeding new rice varieties resistant to weeds.

The advantage of the presented manuscript is the use of original experimental data in the bioinformatics analysis. Another advantage of the presented manuscript is that both new QTLs and QTLs previously found by other authors were obtained during the analysis, which confirms the reliability of the results.

Overall, the article is written clearly, at a high scientific level and describes both the methods used and the results obtained well. The review is quite complete.

However, a serious drawback of the presented manuscript is the significant sloppiness in the design of both individual sections of the text and the figures. The text in some figures is completely illegible. The text of the manuscript contains a significant number of syntax errors. The pdf file submitted for review does not even have line numbers, which makes it extremely difficult to write a review.

Unfortunately, the authors did not analyze SNPs located in the regulatory regions of genes. Such SNPs, localized in the binding sites of transcription factors, can make an important contribution to changes in the expression level of the corresponding genes. It is also not analyzed what substitutions in the amino acid sequences of proteins are caused by non-synonymous SNPs, and where they are localized in the 3-d structure of the protein.

At the same time, the absence of such analysis is not an obstacle to the publication of the article.

Below, specific comments are described. Since the lines of the manuscript are not numbered, for approximate localization of comments it will be necessary to insert pieces of the corresponding text.

  1. «counts, germination rate, coleoptile length, radicle length, total dry weight, were recorded» - Average Dry Weight presented in Supplemental Table 1 is missing.
  2. «SVI = Germination Percentage (%) × Mean Seedling Length (cm)» in Supplemental Table 1, a completely different way of calculating SVI is indicated.
  3. «parents and 564 introgression lines (ILs), 181 lines and their four parental tGBS® SNPs» - clarify on the basis of which criterion exactly 181 of 564 lines were selected.
  4. «(−log p(F) ≥ 4.15)» - indicate why this particular cutoff value was chosen.
  5. «Additional file 1: Sheet 1» - why is GR* TDWG (g) not equal to SVI? For example, for "GSR IR2-6-R2-N1-L2" the difference is 17.06! It is necessary to explain that this discrepancy will not lead to changes in subsequent drawings and does not call into question the conclusions made!
  6. Table 1 needs to be formatted.
  7. Figure 1 - it is desirable to improve the quality of the labels under the axes.
  8. "Weed interference caused reductions in the mean performance of all lines" - and "Leaf Count at 14 DAS" too?
  9. Supplementary Figures 1,2 - it is desirable to add labels directly under the figures.
  10. Figure 2 - it is desirable to improve the quality of the labels under the axes.
  11. Figure 3 - Extremely low quality of the figure, it must be redone. Almost illegible labels and correlation values. In addition, in (A) it is written "Seedling Vigor Index" instead of "Seed Vigor Index". In the figure caption, it is necessary to replace "early seedling vigor (ESG)" with "early seedling vigor (ESV)".
  12. «Principal component analysis among measured traits» - in this section the variance values ​​do not correspond to the values ​​in Figure 4.
  13. Table 3 - move the table to the appropriate section.
  14. «ChengHui448, and Y134. Low-Missing Dataset (LMD50) was filtered, resulting in the» - explain why exactly 50.
  15. Maybe the authors should summarize the results of «SNP markers generated by (tGBS®) sequences for QTL mapping» at the beginning of page 14 in a table?
  16. Page 14. Replace «qSVI2.1 and qSVI2.1» with «qSVI2.1 and qSVI2.2».
  17. Table 4. Explain what is RGP. In addition, it needs to be reformatted. Maybe introduce abbreviations for «polymorphic» and «non-synonymous»?
  18. Page 19. Replace "qSVI2.1 and qSVI2.1" with "qSVI2.1 and qSVI2.2".
  19. For clarity, for the results at the beginning of page 20, it is advisable to make a summary Supplementary table, which will contain a comparison of the obtained results with previously known ones.
Comments on the Quality of English Language

There are syntax errors in the text. The lines of the manuscript are not numbered, and it is difficult for me to refer to these errors.

Author Response

Reviewer 3

Comment 1: The presented manuscript is devoted to the identification of new QTLs in the genomes of breeding lines of rice and genes associated with rice resistance to the weed jungle rice (Echinochloa colona). The authors identify genetic traits that improve early seed germination and seedling vigor in rice, key factors for weed resistance. During the study, conducted on 181 breeding lines of rice, both previously known and new QTLs associated with rice resistance to the weed were identified. In the areas of identified QTLs, genes involved in stress responses were found, in which non-synonymous SNPs were identified. Information about these SNPs can be useful in breeding new rice varieties resistant to weeds.The advantage of the presented manuscript is the use of original experimental data in the bioinformatics analysis. Another advantage of the presented manuscript is that both new QTLs and QTLs previously found by other authors were obtained during the analysis, which confirms the reliability of the results. Overall, the article is written clearly, at a high scientific level and describes both the methods used and the results obtained well. The review is quite complete. However, a serious drawback of the presented manuscript is the significant sloppiness in the design of both individual sections of the text and the figures. The text in some figures is completely illegible. The text of the manuscript contains a significant number of syntax errors. The pdf file submitted for review does not even have line numbers, which makes it extremely difficult to write a review. Unfortunately, the authors did not analyze SNPs located in the regulatory regions of genes. Such SNPs, localized in the binding sites of transcription factors, can make an important contribution to changes in the expression level of the corresponding genes. It is also not analyzed what substitutions in the amino acid sequences of proteins are caused by non-synonymous SNPs, and where they are localized in the 3-d structure of the protein. At the same time, the absence of such analysis is not an obstacle to the publication of the article. Below, specific comments are described. Since the lines of the manuscript are not numbered, for approximate localization of comments it will be necessary to insert pieces of the corresponding text.

Response 1: Thank you for your detailed review and valuable feedback. QTLs and candidate genes for weed tolerance are very minimal, and analyzing such genes requires examining hundreds of them. Therefore, we have left this analysis for future research and researchers' interest. This work will also pave the way for exploring more breeding options to improve rice against weeds.

Comment 2: «counts, germination rate, coleoptile length, radicle length, total dry weight, were recorded» - Average Dry Weight presented in Supplemental Table 1 is missing.

Response 2: We have included in the revised table

Comment 3: «SVI = Germination Percentage (%) × Mean Seedling Length (cm)» in Supplemental Table 1, a completely different way of calculating SVI is indicated.

Response 3: We have corrected in the revised manuscript

Comment 4: «parents and 564 introgression lines (ILs), 181 lines and their four parental tGBS® SNPs» - clarify on the basis of which criterion exactly 181 of 564 lines were selected.

Response 4: Based on the parents' performance in our previous publication, we selected 181 breeding lines derived from these parents for weed competitiveness evaluation and QTL mapping.

Comment 5: «(−log p(F) ≥ 4.15)» - indicate why this particular cutoff value was chosen.

Response 5: In the revised manuscript from line 256-258, the association of a weed competitive trait and a QTL was declared significant once it had a threshold level (−log p(F) ≥ 4.15) based on a permutation test.

Comment 6: «Additional file 1: Sheet 1» - why is GR* TDWG (g) not equal to SVI? For example, for "GSR IR2-6-R2-N1-L2" the difference is 17.06! It is necessary to explain that this discrepancy will not lead to changes in subsequent drawings and does not call into question the conclusions made!

Response 6: Thank you for pointing out this critical error in the supplementary file. This happened because we mistakenly uploaded the wrong file. Previously, we had used the average separately from the two replicates, but in the revised submission, we have corrected this error by properly incorporating both replicates. Additionally, we have used the correct SVI for further analysis.

Comment 7: Table 1 needs to be formatted.

Response 7: We have corrected in the revised manuscript

Comment 8: Figure 1 - it is desirable to improve the quality of the labels under the axes.

Response 8: We have included a separate image file in TIFF format with a resolution of more than 300 dpi. Unfortunately, the image quality is lost when the journal converts it to PDF. We have included the high-quality image in the revised submission

Comment 9: "Weed interference caused reductions in the mean performance of all lines" - and "Leaf Count at 14 DAS" too?

Response 9: No, weed interference did not see significant reductions in "Leaf Count at 14 DAS. However, since we had 181 lines, we have some differences in the introgression lines". We have included a sentence in the revised manuscript. In follow-up work, we can avoid this trait on 14DAS as it doesn’t indicate weed tolerance significantly.

Comment 10: Supplementary Figures 1,2 - it is desirable to add labels directly under the figures.

Response 10: We have incorporated the labels as suggested in the revised submission

Comment 11: Figure 2 - it is desirable to improve the quality of the labels under the axes.

Response 11: We have included a separate image file in TIFF format with a resolution of more than 300 dpi. However, unfortunately, the image quality is lost when converted to PDF by the journal. We have included the high quality image in the revised submission

Comment 12: Figure 3 - Extremely low quality of the figure, it must be redone. Almost illegible labels and correlation values. In addition, in (A) it is written "Seedling Vigor Index" instead of "Seed Vigor Index". In the figure caption, it is necessary to replace "early seedling vigor (ESG)" with "early seedling vigor (ESV)".

Response 12: We have included a separate image file in TIFF format with all the modification

Comment 13: Principal component analysis among measured traits» - in this section the variance values ​​do not correspond to the values ​​in Figure 4.

Response 13: In the revised manuscript, we have clarified the PCA results by explicitly stating the contribution of each principal component. The first two principal components accounted for 63.3% of the total variance, with PC1 explaining 46.9% and PC2 explaining 20.5% for ESG traits, PC1 explaining 50.1%, and PC2 explaining 13.2% for ESV traits.

Comment 14: Table 3 - move the table to the appropriate section.

Response 14: We have corrected in the revised manuscript

Comment 15: «ChengHui448, and Y134. Low-Missing Dataset (LMD50) was filtered, resulting in the» - explain why exactly 50.

Response 15: We have included a brief description of methods of SNP filtrating in the revised manuscript

Comment 16: Maybe the authors should summarize the results of «SNP markers generated by (tGBS®) sequences for QTL mapping» at the beginning of page 14 in a table?

Response 16: A detailed article on SNP markers generated by tGBS® sequences for this population for QTL mapping has already been published. To avoid redundancy and maintain the focus of this manuscript, we have cited the main paper for reference.

Comment 17: Page 14. Replace «qSVI2.1 and qSVI2.1» with «qSVI2.1 and qSVI2.2».

Response 17: We have corrected in the revised manuscript

Comment 18: Table 4. Explain what is RGP. In addition, it needs to be reformatted. Maybe introduce abbreviations for «polymorphic» and «non-synonymous»?

Response 18: We have included in the revised manuscript

Comment 19: Page 19. Replace "qSVI2.1 and qSVI2.1" with "qSVI2.1 and qSVI2.2".

Response 19: We have corrected in the revised manuscript

Comment 20: For clarity, for the results at the beginning of page 20, it is advisable to make a summary Supplementary table, which will contain a comparison of the obtained results with previously known ones.

Response 20: Thank you for the suggestions. As an original research article, we focused on the identified QTLs, using the presence of common QTLs as supporting evidence for their validity. Providing these tables here may undermine our findings. Instead, we plan to summarize all the QTLs identified so far in weed competitiveness in a separate review article currently being prepared.

Reviewer 4 Report

Comments and Suggestions for Authors

1. In the article “Genome-Wide Dissection of Novel QTLs and Genes Associated with Weed Competitiveness in Early-Backcross Selective Regression-Breeding Populations of Rice (Oryza sativa L.)", the authors identify genetic traits that improve early seed germination and seedling vigor in rice, key factors for weed competitiveness.

2. The topic is original and relevant for this field of science. It addresses certain gaps in this area. The data obtained, together with more detailed information, can be used for rice cultivation using direct seed sowing technology. Original to this work is the potential of CRISPR gene expression, cloning, and editing research to improve weed control. By increasing the natural competitiveness of rice against weeds, this research contributes to the sustainable cultivation of rice using the DSR method, reducing dependence on herbicides, increasing crop stability and increasing resource efficiency.

3. These results indicated that using a mapping population of 181 breeding lines derived from Green Super Rice breeding program, revealed 19 quantitative trait loci (QTLs), 17 of which are novel, linked to weed competitiveness. These QTLs, located on eight rice chromosomes, highlight candidate genes involved in stress responses. The findings support marker-assisted and genomic selection for breeding weed-competitive rice varieties.  No research on this topic has been conducted by other authors.  The knowledge gained lays the scientific foundation for the creation of new types of rice varieties.

4. With regard to improving the methodology, the authors should consider the following issues.

In the Methods (Plant Materials) section, the authors write: "The elite GSR breeding population consists of 181 BC1F6 early-backcross selective regression-breeding lines (EB-SILs), which were developed through a cross between the indica rice variety Weed Tolerant Rice1 (WTR1) as the recipient parent, and three donor parents, Haoannong (japonica), ChengHui448 (indica), and Y134 (indica)". However, the WTR1 sample, on the contrary, is a donor of the weed resistance gene, while the rest of the varieties are recipients.

It is not clear why the three populations are mixed together. Genetic research usually examines the segregating population within each crossbreeding combination.

5. The conclusions provide a brief and concentrated summary of the results. However, it is necessary to add the specific most important numerical values from the results.

6. The text in Figures 1-3 is very small and difficult to read. In Figure 4, some words are masked by dots. Figure 5 shows photos of plants with five selected early backcross selective-regression lines, however, photo WTR1-1 is repeated 5 times, Haoannong 2 times, Y134 – 3 times. Redundant information. Table 4 is very cumbersome.

7. References to literary sources are quite appropriate.

Author Response

Reviewer 4

In the article “Genome-Wide Dissection of Novel QTLs and Genes Associated with Weed Competitiveness in Early-Backcross Selective Regression-Breeding Populations of Rice (Oryza sativa L.)", the authors identify genetic traits that improve early seed germination and seedling vigor in rice, key factors for weed competitiveness. The topic is original and relevant for this field of science. It addresses certain gaps in this area. The data obtained, together with more detailed information, can be used for rice cultivation using direct seed sowing technology. Original to this work is the potential of CRISPR gene expression, cloning, and editing research to improve weed control. By increasing the natural competitiveness of rice against weeds, this research contributes to the sustainable cultivation of rice using the DSR method, reducing dependence on herbicides, increasing crop stability and increasing resource efficiency. These results indicated that using a mapping population of 181 breeding lines derived from Green Super Rice breeding program, revealed 19 quantitative trait loci (QTLs), 17 of which are novel, linked to weed competitiveness. These QTLs, located on eight rice chromosomes, highlight candidate genes involved in stress responses. The findings support marker-assisted and genomic selection for breeding weed-competitive rice varieties.  No research on this topic has been conducted by other authors.  The knowledge gained lays the scientific foundation for the creation of new types of rice varieties. With regard to improving the methodology, the authors should consider the following issues.

Comment 1: In the Methods (Plant Materials) section, the authors write: "The elite GSR breeding population consists of 181 BC1F6 early-backcross selective regression-breeding lines (EB-SILs), which were developed through a cross between the indica rice variety Weed Tolerant Rice1 (WTR1) as the recipient parent, and three donor parents, Haoannong (japonica), ChengHui448 (indica), and Y134 (indica)". However, the WTR1 sample, on the contrary, is a donor of the weed resistance gene, while the rest of the varieties are recipients.

Response 1: These are backcross breeding lines that have undergone several rounds of selection under various stress conditions, meaning all the lines are in the WTR1 background with only small favorable introgressions from the donor parents. As seen in the QTL results from Table 3, tolerance alleles contributed from all donor parents, including recipient WTR1. Through stringent positive selection during population development, we have simultaneously improved the population while discovering new gene combinations for stress tolerance.

Comment 2: It is not clear why the three populations are mixed together. Genetic research usually examines the segregating population within each crossbreeding combination.

Response 2: Breeding populations derived from multiple parental lines can be combined to enhance genetic diversity and improve mapping resolution. In our case, the three populations were integrated because they share a common recurrent parent (WTR1) and underwent multiple generations of backcrossing and selection under stress conditions. This approach allows us to capture a broader spectrum of favorable alleles while maintaining the WTR1 background.

Comment 3: The conclusions provide a brief and concentrated summary of the results. However, it is necessary to add the specific most important numerical values from the results.

Response 3: We have included in the revised manuscript

Comment 4: The text in Figures 1-3 is very small and difficult to read. In Figure 4, some words are masked by dots. Figure 5 shows photos of plants with five selected early backcross selective-regression lines, however, photo WTR1-1 is repeated 5 times, Haoannong 2 times, Y134 – 3 times. Redundant information. Table 4 is very cumbersome.

Response 4: We have included a separate image file in TIFF format with a resolution of more than 300 dpi. Unfortunately, the image quality is lost when the journal converts it to PDF. We have included the high-quality image in the revised submission.

These repetitions occur because the best introgression lines are compared to their donor and recipient parents. This comparison is essential to highlight each parent's specific contributions and evaluate the effectiveness of the introgressed segments in improving the target traits.

Comment 5: References to literary sources are quite appropriate.

Response 5: Thank you for your feedback. We appreciate it.

Round 2

Reviewer 1 Report

Comments and Suggestions for Authors Thank you for adding the details. 
Is the sample size 181? In line 19, "181 breeding lines" suggesting 181 types/lines but in line 408 "181 samples" means 181 plants. Please mention the sample size with the text mentioning the statistical tests/p-values

Author Response

Is the sample size 181? In line 19, "181 breeding lines" suggesting 181 types/lines but in line 408 "181 samples" means 181 plants. Please mention the sample size with the text mentioning the statistical tests/p-values.

Thank you for bringing this to our attention. The error occurred due to an inconsistency in terminology during the drafting process. Initially, "samples" and "breeding lines" were used interchangeably, which led to confusion. However, we have now carefully reviewed and corrected this in the revised manuscript, ensuring that "lines" are consistently used throughout the text. Additionally, we have clarified the sample size in sections discussing statistical tests and p-values to avoid further misunderstanding. We appreciate your critical observation.

Reviewer 3 Report

Comments and Suggestions for Authors

The authors took into account most of the comments made by the reviewer. At the same time, a number of important comments remained unaccounted for.

  1. “Comment 6: “Additional file 1: Sheet 1” - why is GR* TDWG (g) not equal to SVI? For example, for "GSR IR2-6-R2-N1-L2" the difference is 17.06! It is necessary to explain that this discrepancy will not lead to changes in subsequent drawings and does not call into question the conclusions made!

Response 6: Thank you for pointing out this critical error in the supplementary file. This happened because we mistakenly uploaded the wrong file. Previously, we had used the average separately from the two replicates, but in the revised submission, we have corrected this error by properly including both replicates. Additionally, we have used the correct SVI for further analysis. »

Reviewer's response. The new «biology-3467609-supplementary.zip» contains the same old «Additional file 1: Sheet 1» with completely incorrect data. It is not clear why the authors are so persistent in trying to use it.

  1. «Comment 12: Figure 3 - Extremely low quality of the figure, it must be redone. Almost illegible labels and correlation values. In addition, in (A) it is written "Seedling Vigor Index" instead of "Seed Vigor Index". In the figure caption, it is necessary to replace "early seedling vigor (ESG)" with "early seedling vigor (ESV)".

Response 12: We have included a separate image file in TIFF format with all the modification»

Reviewer's response. But "early seedling vigor (ESG)" is still not replaced with "early seedling vigor (ESV)" in the figure caption.

  1. “Comment 13: Principal component analysis among measured traits” - in this section the variance values ​​do not correspond to the values ​​in Figure 4.

Response 13: In the revised manuscript, we have clarified the PCA results by explicitly stating the contribution of each principal component. The first two principal components accounted for 63.3% of the total variance, with PC1 explaining 46.9% and PC2 explaining 20.5% for ESG traits, PC1 explaining 50.1%, and PC2 explaining 13.2% for ESV traits.”

Reviewer's response. But in Figure 4A for ESG traits, PC1 explains 50.1% (not 46.9%) and PC2 explains 13.2% (not 20.5%) of the variance. Similarly, for ESV traits, PC1 explains 42.8% (not 50.1%) and PC2 explains 20.5% (not 13.2%) of the variance (Figure 4-B). It is obvious that the authors mixed up the figure captions. Moreover, it is not clear where the value 46.9% came from in the text of the manuscript. This value is not shown in Figure 4.

This attitude to data analysis, along with the use of incorrect source data in "Additional file 1: Sheet 1", raises questions about the reliability of all the work done within the article.

  1. "Comment 14: Table 3 - move the table to the appropriate section.

Response 14: We have corrected in the revised manuscript"

Reviewer's response. Table 3 remained in the same place, but the reference to Table 3 completely disappeared from the text of the manuscript. It now needs to be added to the text.

Author Response

  1. “Comment 6: “Additional file 1: Sheet 1” - why is GR* TDWG (g) not equal to SVI? For example, for "GSR IR2-6-R2-N1-L2" the difference is 17.06! It is necessary to explain that this discrepancy will not lead to changes in subsequent drawings and does not call into question the conclusions made!

Response 6: Thank you for pointing out this critical error in the supplementary file. This happened because we mistakenly uploaded the wrong file. Previously, we had used the average separately from the two replicates, but in the revised submission, we have corrected this error by properly including both replicates. Additionally, we have used the correct SVI for further analysis. »

Reviewer's response. The new «biology-3467609-supplementary.zip» contains the same old «Additional file 1: Sheet 1» with completely incorrect data. It is not clear why the authors are so persistent in trying to use it.

We sincerely apologize for the repeated mistake in uploading the correct supplementary file. We did not intend to persist with incorrect data, and we genuinely appreciate your critical observation in pointing this out. We carefully verify and ensure that the correct version is included in our current submission. If any concerns remain, we will provide the relevant images and raw data to help clear up this discrepancy. Thank you for being so understanding.

  1. «Comment 12: Figure 3 - Extremely low quality of the figure, it must be redone. Almost illegible labels and correlation values. In addition, in (A) it is written "Seedling Vigor Index" instead of "Seed Vigor Index". In the figure caption, it is necessary to replace "early seedling vigor (ESG)" with "early seedling vigor (ESV)".

Response 12: We have included a separate image file in TIFF format with all the modification»

Reviewer's response. But "early seedling vigor (ESG)" is still not replaced with "early seedling vigor (ESV)" in the figure caption.

We have now corrected this in the revised submission by replacing "early seedling vigor (ESG)" with "early seedling vigor (ESV)" in the figure caption.

  1. “Comment 13: Principal component analysis among measured traits” - in this section the variance values ​​do not correspond to the values ​​in Figure 4.

Response 13: In the revised manuscript, we have clarified the PCA results by explicitly stating the contribution of each principal component. The first two principal components accounted for 63.3% of the total variance, with PC1 explaining 46.9% and PC2 explaining 20.5% for ESG traits, PC1 explaining 50.1%, and PC2 explaining 13.2% for ESV traits.”

Reviewer's response. But in Figure 4A for ESG traits, PC1 explains 50.1% (not 46.9%) and PC2 explains 13.2% (not 20.5%) of the variance. Similarly, for ESV traits, PC1 explains 42.8% (not 50.1%) and PC2 explains 20.5% (not 13.2%) of the variance (Figure 4-B). It is obvious that the authors mixed up the figure captions. Moreover, it is not clear where the value 46.9% came from in the text of the manuscript. This value is not shown in Figure 4.

This attitude to data analysis, along with the use of incorrect source data in "Additional file 1: Sheet 1", raises questions about the reliability of all the work done within the article.

We sincerely appreciate the reviewer's thorough analysis and for pointing out these discrepancies. We regret the oversight and any confusion it may have caused. In the final revised version, we have carefully verified and corrected the variance values to ensure they accurately correspond to Figure 4. Also, please look at Supplementary Figure 2, where we also PCA to Loading plot showing the vector coefficients. Additionally, we have double-checked all data sources to ensure consistency and reliability across the manuscript. We would happily provide the relevant images and raw data for further clarification if there are any remaining concerns. We genuinely appreciate your critical comments for improving the overall manuscript quality.

  1. "Comment 14: Table 3 - move the table to the appropriate section.

Response 14: We have corrected in the revised manuscript"

Reviewer's response. Table 3 remained in the same place, but the reference to Table 3 completely disappeared from the text of the manuscript. It now needs to be added to the text.

The MDPI publisher's automated software mistakenly assigned Table 3 to the wrong place, removing the reference from the text. We have already requested the editor share the raw files with you for clarification. Similarly, the poor image quality from the beginning is due to the automated system extracting images from the text rather than using the separately submitted high-resolution files. We appreciate your patience and understanding as we work with the publisher to resolve these issues.